# Targeting Mitochondria to Control Ageing and Senescence

**DOI:** 10.3390/pharmaceutics15020352

**Published:** 2023-01-20

**Authors:** Margherita Protasoni, Manuel Serrano

**Affiliations:** 1Institute for Research in Biomedicine (IRB Barcelona), Barcelona Institute of Science and Technology (BIST), 08028 Barcelona, Spain; 2Catalan Institution for Research and Advanced Studies (ICREA), 08010 Barcelona, Spain; 3Cambridge Institute of Science, Altos Labs, Granta Park, Cambridge CB21 6GP, UK

**Keywords:** ageing, cellular senescence, mitochondrial targeting, novel therapeutic approaches

## Abstract

Ageing is accompanied by a progressive impairment of cellular function and a systemic deterioration of tissues and organs, resulting in increased vulnerability to multiple diseases. Here, we review the interplay between two hallmarks of ageing, namely, mitochondrial dysfunction and cellular senescence. The targeting of specific mitochondrial features in senescent cells has the potential of delaying or even reverting the ageing process. A deeper and more comprehensive understanding of mitochondrial biology in senescent cells is necessary to effectively face this challenge. Here, we discuss the main alterations in mitochondrial functions and structure in both ageing and cellular senescence, highlighting the differences and similarities between the two processes. Moreover, we describe the treatments available to target these pathways and speculate on possible future directions of anti-ageing and anti-senescence therapies targeting mitochondria.

## 1. Introduction

Cellular senescence is recognized as a hallmark of ageing [1] and cancer [2]. Cellular senescence is a response to severe damage or stress characterised by the inability to proliferate and by a robust secretion of high amounts of inflammatory, fibrogenic and mitogenic factors, collectively known as the senescence-associated secretory phenotype (SASP) [3,4]. Cellular senescence serves important physiological functions, most notably to initiate tissue repair and to prevent the outgrowth of potentially oncogenic cells. After exerting their beneficial functions, senescent cells are normally cleared by the immune system. However, with ageing, senescent cells are not efficiently cleared and accumulate contributing to many diseases. Indeed, the accumulation of senescent cells can lead to disruption of tissue functionality, and limit the regenerative potential of adult stem cells by damaging the stem cell niches within the affected tissue [5]. Although still speculative, the SASP, rather than the physical presence of senescent cells, could be the main pathological agent of cellular senescence.

For the purposes of this review, we will use the terms “cellular senescence” and “cellular ageing” to describe two different cellular processes. We will use “cellular ageing” when referring to what happens to cells when the organism ages, and we will use “cellular senescence” when referring to what happens to cells when they reach a threshold of severe damage and undergo adaptations that profoundly change cellular biology. “Cellular ageing” involves progressive changes that deteriorate cell functions, often over the span of years, but without drastically changing their main biological properties. In contrast, “cellular senescence” involves a dramatic transformation of the cell biology in a process that typically takes about 7–10 days (summarised in Figure 1). While “aged cells” are suboptimal versions of their younger counterparts, “senescent cells” are very different versions with multiple aspects of their biology altered, including major chromatin reconfiguration, remarkable expansion of the lysosomal compartment, and increased autophagy [4]. In the following sections, we will try to summarise and compare how mitochondrial biology changes during “cellular ageing” and upon “cellular senescence”. 

There is a growing interest in therapeutically targeting both “aged cells” and “senescent cells”. The goals are different: in the case of “cellular ageing”, it would be ideal to find interventions that (1) slow down the pace of deterioration of cells, or (2) reverse the accumulated damage. In the case of “cellular senescence”, the goals are (1) to eliminate senescent cells either by driving them into apoptosis (senolytic strategies) or by stimulating their immuneclearance, or (2) to reduce the production of their main pathological mediator, namely, the SASP. It is important to emphasise that the senescence-targeting therapies do not prevent or interfere with the de novo implementation of senescence (which is an important anti-oncogenic barrier), but act on pre-existing senescent cells that have not been naturally cleared by the immune system. In this review, we will discuss strategies particularly targeted to the mitochondrial biology of “cellular ageing” and “cellular senescence”.

Mitochondria are ubiquitous intracellular organelles essential for multiple cellular functions. Indeed, these organelles are central in the metabolic processes involved in ATP and energy production but are also implicated in calcium homeostasis, ROS signalling, apoptosis, haem and iron-sulphur clusters synthesis, inflammation, and epigenetics regulation. Numerous studies have shown how mitochondria in aged and senescent cells undergo heavy structural changes and general functional decline (summarised in Figure 2), suggesting that this organelle might be a favourable target to tackle ageing-associated diseases. However, the understanding of these modifications is still limited and additional and improved experimental and clinical data is required before applying this knowledge to the healthcare practice. 

## 2. Mitochondria as Inflammation Triggers

Ageing is associated with increased inflammation and activation of the innate immune system. This condition is known as “inflamm-ageing” and is characterised by chronic activation of JAK-STAT signalling in the circulating immune cells of elderly patients [6], activation of the NLRP3 inflammasome [7,8,9], and higher circulating levels of inflammatory mediators such as C-reactive protein, IL-6, and fibrinogen [10]. A leading hypothesis for the origin of “inflamm-ageing” is the build-up of senescent cells with ageing. An important support for this hypothesis comes from experiments in which aged mouse blood is transferred to young animals, which results in features of accelerated ageing. Interestingly, previous treatment of the old donors with senolytic agents reduced “inflamm-ageing” after blood exchange, and the old blood lost its pro-ageing activity [11]. In humans, senolytic treatments also reduce the “inflamm-ageing” of patients suffering from lung fibrosis [12] or chronic kidney disease [13]. 

Importantly, mitochondria of senescent cells are known to play a key role in triggering the SASP. In particular, depriving senescent cells of mitochondrial DNA [14] or mitochondria altogether [15] seriously compromises the SASP. The detailed mechanisms connecting the mitochondria of senescent cells with the SASP are still unknown. We speculate that they could be similar to the mechanisms connecting dysfunctional mitochondria with inflammation [16]. These may include the release of cytosolic and/or extracellular mitochondrial DNA (mtDNA), mitochondrial double-stranded RNA, *N*-formyl peptides (a sub-product of mitochondrial protein translation), and phospholipid species such as cardiolipin, enriched in the inner mitochondrial membrane (IMM) [17,18,19]. The most studied of these components is mtDNA which will be analysed in depth in the next section. Apart from mtDNA, it is worth mentioning that formyl peptides can be released extracellularly by damaged mitochondria and activate neutrophils by engaging their formyl peptide receptor-1 (FPR1) [18]. Specific FPR1 antagonists have been generated and used to treat brain tumours and neurodegeneration [20,21], suggesting a potential use to fight age-driven inflammation. 

## 3. Mitochondrial DNA

As a reflection of their bacterial origin, mitochondria contain their own genetic material, mtDNA. This circular double-stranded molecule only counts 16,569 base pairs [22], but it is present in hundreds of copies in each cell and its contribution to organismal ageing has been extensively discussed. The initial hypothesis was that the accumulation of mtDNA mutations with age might directly contribute to the decline of mitochondrial functions. Indeed, compared to nuclear DNA, mtDNA is in close contact with the electron transport chain, the principal producer of reactive oxygen species (ROS) that can induce oxidative DNA damage, is less protected, and its repair mechanisms are far less sophisticated [23]. Supporting this hypothesis, mtDNA deletions and mutations are detected in tissues from aged animals and humans [24,25,26,27]. Moreover, the generation of an mtDNA mutator mouse model, which expresses a proof-reading-deficient version of the mitochondrial DNA polymerase G and accumulates mtDNA mutations at vastly increased rates, showed reduced lifespan and premature onset of ageing when the mutation is in homozygosity [27,28]. However, heterozygous DNA polymerase G mutant mice show normal ageing despite huge levels of mtDNA mutations and the amount of mtDNA mutations that accumulate during natural ageing is far lower than in these mutant mice [29]. As is true for oncogenic nuclear DNA mutations, it has been shown that mtDNA mutations accumulate in several human tumours, particularly in genes encoding for complex I subunits. These mutations favour oncogenesis by inducing metabolic remodelling, accelerating cell proliferation, and reducing apoptosis at least in certain tissues [30,31]. In this case, cellular senescence could be initiated as a defence mechanism to suppress the development of a tumour [23]. 

Less controversial, instead, is the role of mtDNA on organismal ageing when released outside of the mitochondrial matrix, both into the cytosol or extracellularly. This process increases in senescent cells [32] and is now a hot topic of research because of its impact on inflammation and immune responses due to the mitochondrial genome’s bacterial-like nature. Cytosolic escape of mtDNA triggers the cGAS-STING-NLRP3 axis, a key process in response to cellular stress [33,34], and consequently activates the interferon regulatory factor 3 (IRF3) or the transcription factor family nuclear factor kB (NF-κB) pathway, major players in inflammation and antiviral response [35,36,37]. The mechanisms involved in mtDNA escape are still debated, which makes finding direct inhibitors complicated. An indirect strategy has been to reduce mtDNA release as a downstream effect of other treatments including reducing oxidative stress with melatonin supplementations [38], or reducing the opening of the mitochondria permeability transition pore (mPTP), a possible exit way for mtDNA outside of the matrix, with cyclosporin A [39]. These studies, however, have not yet offered a clear mechanistic explanation of how these compounds inhibit mtDNA release. Another approach is to act directly on the cGAS pathway. A variety of small molecules able to inhibit cGAS activation have been successfully identified and developed in treating autoimmune diseases, such as RU.521, which competitively binds to cGAS catalytic pocket with cGAS substrates ATP/GTP [40], Cu-32 and Cu-76, that prevent cGAS dimerisation and subsequence activation [41], and additional small molecules identified through screenings: G140/G150 [42] and PF-06928215 [43]. When mtDNA is released in the extracellular environment and reaches the bloodstream, instead, it can be taken up by immune cells, such as neutrophils and macrophages, by endocytosis, and activate the Toll-like receptor 9 (TLR9), a pillar of antibacterial and antiviral responses [44,45]. The downstream effect is the activation of NF-κB, the secretion of tumour necrosis factor-ɑ (TNF-ɑ), and the expression of the pro-interleukin-1β and -18 in both tissue-resident macrophages and circulating leukocytes [46,47], leading to the recruitment of other immune factors. Circulating mtDNA appears to increase gradually with age after the fifth decade of life and to be strictly associated with inflammatory status [48]. Indeed, levels of circulating mtDNA correlate with serum inflammatory markers [48]. For this reason, designing new therapeutic strategies against circulating mtDNA, or the receptors it binds, could be relevant in the future treatment of ageing-associated diseases. 

## 4. Mitochondrial Life: Biogenesis, Dynamics and Mitophagy

Mitochondria are very dynamic organelles. They undergo constant fusion and fission events to create a specific network able to accommodate the cellular energy demands and metabolic state, allow transport, and favour the selective removal of damaged mitochondria through a process known as “mitophagy” [49]. While elongated mitochondria permit the sharing of metabolites, proteins, and mtDNA and enhance cell survival, mitochondrial fragmentation is often associated with motility or mitophagy, and, in more extreme cases, is a sign of mitochondrial dysfunction and cell death [50].

Mitochondrial dynamics change with organismal ageing and cellular senescence. Ageing in flies and mammalians is characterised by enlarged mitochondria, irregular cristae shape and size, and a decrease in mitochondrial number [51,52]. In addition, senescent cells, are characterised by a very elongated and branched mitochondrial network [53,54]. A possible cause for this fusion-oriented phenotype is the reduced expression of FIS1 and DRP1, two of the proteins involved in promoting mitochondrial fission, during senescence [55]. This downregulation might reflect an attempt to dilute and re-arrange matrix content between healthy and damaged organelles or to resist apoptosis [56]. Indeed, there is a tight interconnection between apoptosis and mitochondrial dynamics since DRP1 relocates from the cytosol to mitochondria during cell death, resulting in mitochondrial fragmentation, loss of membrane potential, and cytochrome *c* release [57,58]. When this protein’s activity is inhibited, the development of a senescent phenotype is favoured [59], while the induction of Drp1p expression in Drosophila midlife prolongs both life- and health-span via improved mitochondrial respiration and autophagy [60]. Similarly, the induction of mitochondrial fission in the intestine of both *C. elegans* and flies increases longevity [61]. Upregulation of mitochondrial fission could therefore ameliorate senescence-related phenotypes. 

Another common hallmark of organismal ageing and cellular senescence is impaired mitophagy [62,63], which leads to the accumulation of dysfunctional organelles, as observed in both old rats and humans cells [64,65], and also in senescent cells in vitro and in vivo [66,67]. Mitophagy decline might result from several molecular mechanisms. Firstly, defects in lysosomal function or lysosomal overload might prevent lysosomal enzymes from targeting autophagosomes, leading to defective removal of dysfunctional mitochondria [61,68]. In aged cells from old tissues as well as in senescent cells, lysosomes show reduced activity and accumulation of undegraded material [69]. Secondly, an overall defect in cellular autophagic capacities could explain the deficiencies in mitochondrial clearance. The mTOR-autophagy axis is affected during ageing, and senescent cells show elevated mTORC1 activity [15,66,70], becoming unresponsive to starvation signals [71]. This hypothesis is supported by studies where autophagic flux was restored and age-related conditions were prevented in rodents, dogs, non-human primates, and humans after treatment with two mTOR inhibitors: rapamycin [72,73,74] and AZD80055 [75]. In addition, impaired mitophagy might derive from the mitochondrial dynamics defects previously described. Indeed, the efficiency of this process relies on the ability of the organelle to undergo fission and segregate the segment of the network that needs to be eliminated [76]. Moreover, both fission and mitophagy efficiency can be worsened with ageing because of a reduced expression of PINK1 [77], which, together with Parkin, is the main actor of the mitochondrial clearance pathway [78].

While mitochondrial turnover decreases in both aged and senescent cells, mitochondrial biogenesis slows down during ageing [79] but appears to increase during senescence, leading to a rise in respiration and ROS production [80]. The underlying reasons still have to be completely elucidated, but a possible explanation for this difference is the expression level of a key regulator of biogenesis, PGC-1α, which is higher in senescent cells [80] and reduced in aged animals [81]. However, despite different screenings to identify compounds able to modulate PGC-1α transcription, results are still elusive [82,83] and alternative approaches are needed. Another proposed way to pharmacologically modulate mitochondrial biogenesis is targeting the AMP-activated protein kinase (AMPK) pathway. In aged animals, chronic AMPK inactivation is associated with a marked decrease in mitochondrial biogenesis [84], while reduced AMPK activity correlates with ageing-related insulin resistance and insufficient intracellular fat oxidation [85]. On the other hand, when aak-2, the worm equivalent of AMPK, is overexpressed in *C. elegans*, it results in an increased lifespan [86]. Thus, chronic activation of AMPK via metformin treatment, a compound already used in the clinic for the treatment of type 2 diabetes, has been proposed as a strategy for slowing ageing [87]. However, while preliminary studies showed promising results, it is unclear if the benefits come from its direct action on AMPK or indirect effects on cellular metabolism and glycaemia, reduction of oxidative stress, or protective effects on the endothelium and vascular function [88]. The molecular mechanism of metformin is also debated, since metformin can activate AMPK as a downstream effect of its interaction with various proteins and pathways, including mitochondrial complex I, the nuclear receptor NR4A1, and the endosomal/lysosomal v-ATPase [89,90]. Moreover, the effects of metformin on mice longevity are not robust across different laboratories. While certain studies showed increased lifespan [91], others reported no changes [92] or even decreased longevity [93], highlighting the importance of comprehensive testing before the introduction of metformin-based anti-ageing therapies. 

Interestingly, the role of AMPK activity and its function in the regulation of mitochondrial biogenesis in cellular senescence is less clear. A study in immortalised human fibroblasts expressing SV40 large T antigen showed that inactivation of large T resulted in an increase in AMPK activity that directly contributed to the implementation of senescence [94]. In contrast, a study in H_2_O_2_-induced senescent murine fibroblasts found AMPK inactivated [95]. Since AMPK is central in multiple cellular pathways, additional research could clarify the observed differences between models, and unravel additional molecular mechanisms involved in the establishment and/or maintenance of senescence. 

## 5. Mitochondrial Unfolded Protein Response

Mild mitochondrial stress can be beneficial for longevity [96,97] and a reason for this hormetic effect could be the activation of the mitochondrial unfolded protein response (UPRmt) [98]. The UPRmt, similarly to the endoplasmic reticulum unfolded protein response (UPR_ER_) and the cytoplasmic heat shock response (HSR), is capable of initiating a broad-range transcriptional response that not only is involved in the refolding of mitochondrial matrix proteins, but also in ROS defences, metabolic changes, regulation of iron-sulphur cluster assembly, and modulation of the innate immune response [99,100,101]. Lead UPR factors include the heat shock protein 60 and 10 (HSP60 and HSP10), mitochondrial heat shock protein 70 (mtHSP70), Lon peptidase 1 (LONP1) and caseinolytic protease (ClpP).

While it has not been clearly described if, when, and how a decline in UPRmt functionality takes place during the ageing of the organism, it is largely accepted that UPRmt activation has a beneficial effect on longevity since it promotes cell survival and the recovery of the mitochondrial network and cellular function. However, the beneficial activation of UPRmt might not be a viable route in tackling ageing therapeutically. Indeed, UPRmt triggering after exposure to mitochondrial stress shows strong responses only during development [102,103], while it appears less active in later stages of life when there is no reported increased lifespan as a response to stressors [96,104,105]. This timing limitation has been justified by the fact that mitochondrial perturbations early in life induce widespread changes in chromatin structure through histone H3K9 di-methylation and long-lasting effects on gene expression [106]. The transcription of target UPRmt genes is subject to epigenetic regulation by histone3-specific methylation and is therefore influenced by those stresses that occurred during development [106,107] while being less sensitive to treatments in aged organisms. 

Similarly, limited experimental data are available about the UPRmt and senescence. In senescent hepatocytes, most of the UPRmt factors levels were found significantly reduced and the pathway compromised [108], suggesting that UPRmt targeting could have a role in preventing senescence. Similarly, experiments in senescent human lung fibroblasts showed a reduced ability to cope with the accumulation of mitochondrial unfolded proteins [109]. Additional research is needed to investigate this hypothesis. 

## 6. Metabolism and Electron Transport Chain

Mitochondria’s primary function is to be the “powerhouse of the cell”. Thanks to the activity of the electron transport chain (ETC, composed of four enzymatic complexes embedded in the IMM), ATP is formed from adenosine diphosphate and inorganic phosphate and becomes available for the cell to be used as “energy currency” [110]. Analyses of mitochondrial function in skeletal muscle samples from older subjects showed a strong decline in mitochondrial respiratory capacity and a reduction in ATP amount [111,112]. At the same time, ROS produced by the ETC enzymes increase in aged animals [24]. Similarly, while experiments on senescent fibroblasts showed an increase in mitochondrial mass and abundance of tricarboxylic acid (TCA) cycle metabolites [113,114], the efficiency of the ETC and ATP production appeared reduced, leading to decreased mitochondrial membrane potential, increased proton leak and generation of ROS [115,116]. Because of this defect, senescent cells appear to undergo a metabolic switch, increasing dependence on glycolysis [117], and fatty acid oxidation [118]. As discussed previously, the cause of the ETC damage during both ageing and the establishment of cellular senescence is probably the combination of increased ETC machinery malfunction due to progressive damage and a decline in the removal of dysfunctional mitochondria.

For all these reasons, a direct or indirect amelioration of the energetic capacity and functionality of mitochondria in ageing and senescent cells could improve patients’ symptomatology. This result could be achieved by reducing stressful or damaging conditions, such as ROS and calcium accumulation, impaired mitophagy or altered dynamics. However, the ETC itself can be directly targeted. For example, both aged tissue and senescent cell exhibit low levels of Coenzyme Q_10_ (CoQ_10_), the ETC carrier that collects electrons from complexes I and II and delivers them to complex III [119,120]. This deficiency can lead to electron leakage, loss of membrane potential, ROS production, and reduced ETC efficiency [121]. In mice with accelerated ageing, CoQ_10_ supplementation improves complexes I and IV activity and OXPHOS efficiency and decreases ROS generation, slowing down the progression of ageing-related symptoms and preventing ageing [122]. CoQ_10_ supplementation could have beneficial effects also against the development of cellular senescence, as demonstrated in mesenchymal stem cells [123] and H_2_O_2_-induced senescent HUVECs [124].

Furthermore, favouring alternative energy-producing pathways such as β-oxidation can show beneficial effects on lifespan and metabolism [125]. Direct administration of fatty acids, including α-linolenic or omega-3 fatty acid, indeed, promotes higher mitochondrial energy production, mitochondrial biogenesis, and oxidative stress reduction [126]. This hypothesis was tested also in the clinic, where humans aged 65 to 85 showed increased mitochondrial protein synthesis and significantly reduced mitochondrial oxidative stress after being treated with ω-3 polyunsaturated fatty acids [127]. Another proposed treatment able to increase β-oxidation is the supplementation of 17α-estradiol, a weak endogenous steroidal oestrogen. 17α-estradiol improved metabolic parameters and slowed ageing in male mice, but did not show significant effects in females [128]. However, conclusive clinical trials about the impact of these treatments on human health and longevity are still missing.

## 7. NAD^+^ Levels

Nicotinamide adenine dinucleotide (NAD^+^) is one of the most common metabolites in the human body and an indispensable cofactor involved in several redox reactions. Most NAD^+^ functions as a redox carrier, receiving electrons from metabolic processes such as glycolysis, Krebs cycle and β-oxidation, and forming NADH. NADH is then used to transfer electrons to complex I in the ETC [129]. Approximately 10% of cellular NAD^+^, instead, can be phosphorylated by NAD^+^ kinases into NADP^+^ and used for anabolic reactions, such as lipid and nucleic acid syntheses, which require NADPH as an electron donor [130,131].

NAD^+^ levels decline with age in various tissues, and this reduction correlates with the development of ageing-associated diseases such as muscle loss and diabetes [132,133,134]. A drop in NAD^+^ levels, indeed, associates with mitochondrial dysfunction in both calorie-rich diets and ageing, whereas NAD^+^ repletion with precursors such as nicotinamide riboside and nicotinamide mononucleotide can reverse this process, improving mitochondrial respiration and increasing ETC subunits expression [132,133,135]. Nonetheless, decreased NAD^+^/NADH ratios or total NAD^+^ levels can drive senescence and cell cycle arrest, but also influence the SASP phenotype [14,136]. For example, cells that underwent mitochondrial dysfunction-associated senescence are characterised by lower NAD+/NADH ratios, AMPK-mediated p53 activation, and reduced IL-1-associated SASP [14]. The complexity of the role of NAD^+^ in the establishment of senescence is linked to its activity as a cofactor of two important protein families: poly-ADP-ribose polymerase (PARP) and sirtuin family proteins (SIRTs). PARP has a double and contradictory role in the establishment of senescence, making it difficult to use this protein as an anti-ageing target. On one side, indeed, PARP inhibition or depletion leads to single-strand breaks, cell cycle arrest, and cellular senescence [137,138,139]. On the other side, its activity promotes NF-kB activation and secretory phenotypes in senescent cells [140]. Consistently, a decrease in NAD^+^ amount by inhibition of Nicotinamide phosphoribosyltransferase, responsible for the NAD salvage pathway, promotes both cell cycle arrest and suppresses SASP [14,136]. While all the mentioned studies have been conducted on human cells, the use of different cell models (keratinocytes, ovarian cancer, colorectal cancer, melanoma and breast cancer cells) and the use of different senescence triggers might explain these disparities. Likewise, SIRTs play a role in senescence and SASP generation. The mitochondrial sirtuin SIRT3 is critical in the elimination of intracellular ROS and the maintenance of oxygen metabolism balance [141]. Interestingly, the increased oxidative stress that results from its depletion can induce different senescence markers but suppresses SASP secretion. Finally, it has been recently discovered that NAD^+^ levels can be restored in stressed or damaged cells by a cytosolic complex of enzymes that transfers electrons from NADH to NADP+, and this reaction can prevent cellular senescence [142]. 

For all these reasons, the modulation of NAD^+^ levels and the downstream molecular pathways have been largely studied as a potential target for anti-ageing therapies. The most direct proposed intervention was to increase NAD^+^ levels by dietary, via supplementation of NAD^+^ precursors, such as nicotinic acid, nicotinamide riboside, nicotinamide mononucleotide, and tryptophan, or improving NAD^+^ bioavailability through exercise and caloric restriction. However, while these therapies have already shown promising results in clinical trials, it is necessary to remember that NAD^+^-depleting drugs have an anti-tumoural effect, and the long-term boosting of NAD^+^ might increase the risk of developing cancer. Consistent with this observation, nicotinamide mononucleotide treatment accelerates pancreatic cancer progression inducing an inflammatory environment [136]. Another strategy could be the reduction of PARP1 and CD38 activity, which consume NAD^+^ [143,144]. US-FDA-approved PARP inhibitors, such as niraparib, rucaparib and olaparib, are already available and used to treat cancers, including prostate, breast and ovarian, through disrupting DNA repair and replication pathways [145,146,147]. Similarly, CD38 inhibitors have been proposed, such as apigenin, quercetin, luteolin, kuromanin, luteolinidin, and 78c [148]. 78c, for example, is a highly specific CD38 inhibitor which showed promising results in reversing NAD^+^ decline during ageing and improving age-associated cardiac and muscle function and glucose tolerance [143]. The opposite tactic, instead, could be stabilising NAD^+^ levels by increasing NAD^+^-biosynthesis enzymes activity [149] or preventing the escape of intermediates [150].

Finally, SIRTs themselves could represent a powerful tool in anti-ageing therapies. Overexpression of SIRT3 showed beneficial effects on ageing and senescence hallmarks [151] and significantly activates mitophagy [152]. Increased SIRT3 activity can be reached through calorie restriction [153] or compounds such as adjudin, a derivative of lonidamine [154], which was described attenuating cellular senescence markers in hydroxyurea-treated MEFs [155]. However, the potential of this pharmacological approach remains to be validated in clinical conditions. 

## 8. Matrix Calcium

Calcium (Ca^+2^) is an ion that participates in a wide variety of cellular functions, being an intracellular regulator of many physiological processes. Mitochondria and Ca^+2^ are strictly interconnected. On one hand, the cell benefits from the mitochondrial ability to buffer Ca^2+^, shaping the cytosolic Ca^2+^ signal [156,157]. On the other, Ca^2+^ is fundamental for normal mitochondrial functions since it activates pyruvate, isocitrate, and 2-oxoglutarate dehydrogenases, involved in the TCA cycle, stimulating mitochondrial respiration and ATP production [158]. Excessive and prolonged accumulation of mitochondrial Ca^2+^, often in combination with increased production of ROS [159], however, can be toxic and regulate cell death through the stable opening of the mPTP [160]. This phenomenon leads to mitochondrial swelling, metabolism impairment, alterations in the matrix content, membrane potential drop, and apoptosis [161,162,163]^.^ Because of this, calcium movements within organelles are dynamic but strictly controlled. Its route starts in the endoplasmic reticulum (ER), which releases calcium through ryanodine receptors and inositol 1,4,5-triphosphate receptors (Ins(1,4,5)P3Rs). In particular, Ins(1,4,5)P3Rs are enriched at the ER-mitochondria contact sites, areas of proximity but not fusion, between the membranes of the two organelles [164], and are activated by IP3 binding. On the receiving side, mitochondria express VDAC1 in the outer mitochondrial membrane (OMM) and the mitochondrial calcium uniporter complex (MCUC) in the IMM that uptakes Ca^2+^ into the matrix.

Recent observations pointed out that a rise in intracellular Ca^2+^ contents could be a new hallmark of ageing and cellular senescence, both at a cytosolic level and in the intracellular organelles [165,166], leading to chronic mitochondrial stress and cellular toxicity. Evidence shows increased activation of Ins(1,4,5)P3Rs in the ER membrane of senescent cells and consequent amplified uptake of Ca^2+^ through MCU channels, possibly due to a decreased expression of the transient receptor potential cation channel subfamily C member 3 (TRPC3) [167], a controller of mitochondrial Ca^2+^ load. Re-expression of TRPC3, indeed, diminished mitochondrial Ca^2+^ load of these cells and promoted escape from oncogene-induced senescence. Nonetheless, additional alterations in the mitochondrial calcium machinery structure or activity cannot be excluded and should be further investigated. 

Mitochondrial calcium concentrations could be, therefore, an attractive target in both anti-ageing and anti-senescence therapies. While calcium modulation in senescent cells is still poorly studied, it has been proposed to use specific MCU inhibitors, such as Ru360 or Ru265, to reduce the entry of Ca^2+^ in the matrix of ageing cells, and consequentially mitochondrial stress. These compounds have been largely characterised in vitro and in vivo, and have shown promising protective activity in reperfusion [168] and hypoxic/ischemic brain injury [169] animal models. Recent work in *C. elegans* started testing this hypothesis in the context of ageing and reported that pharmacological or genetic inhibition of MCU was sufficient to improve muscle ageing and dystrophy, corroborating this hypothesis [170]. In addition, future studies should investigate other possible therapeutic targets, including the modulation of ER-mitochondria tethering and Ins(1,4,5)P3Rs activity, the role of other subunits of the MCU complex, and other mechanisms of mitochondrial calcium influx and efflux in ageing and senescence.

## 9. Reactive Oxygen Species

Mitochondria are a major source of ROS, which are primarily the result of the inefficient transfer of electrons through the ETC. According to the mitochondrial free radical theory of ageing, ROS are both a central cause and a consequence of ageing. Indeed, reportedly they increase with age because of a decline in ETC capacity, respiratory complexes dysfunctions, and a decrease in ROS scavenging enzymes, whereas, on the other side, ROS accumulation increases levels of oxidized lipids and proteins, induces mtDNA mutations, and further deteriorates the ETC [171,172,173]. In cellular senescence, a similar pattern can be observed: ROS contribute to cellular senescence onset inducing oxidative damages [174] and inhibiting autophagy [175], while the chronic accumulation of these species establishes a vicious cycle of mitochondrial and cellular stress.

Thereby, it would be easy to believe that the elimination or drastic reduction of ROS would be an effective anti-age therapy. While this reasoning is not intrinsically wrong, the situation is more complicated. Mitochondria-targeted antioxidant drugs such as plastoquinone derivatives [176,177] or MitoTEMPO [178,179], and endogenous indoleamine melatonin [180], validate this theory, showing increased lifespans in mice and flies and generally improved mitochondrial functions. Likewise, genetically modified mice with reduced ROS production show delayed ageing [181,182]. On the contrary, however, additional reports raised doubts about the free radical theory of ageing and the use of antioxidant therapies. In mice, the overexpression of major antioxidant enzymes such as copper-zinc superoxide dismutase (CuZnSOD or SOD1), catalase, or manganese superoxide dismutase (MnSOD) did not increase longevity [183], while deletion of mitochondrial matrix SOD increased mtDNA damage and cancer incidence but did not accelerate ageing [184,185]. In *C. elegans,* instead, loss of superoxide dismutase enzymes could even extend lifespan [186]. Finally, the use of certain antioxidants on proliferating cells can favour cellular senescence, by inducing proliferation arrest, DNA damage and chromosomal abnormalities [187]. 

These controversies highlight that cellular ROS are not only a damaging “waste product” of mitochondrial activity but have critical functions in cellular life. Emerging evidence, indeed, show the importance of ROS in cellular signalling. For example, H_2_O_2_ generated from superoxide produced by mitochondria and NADPH oxidases [188,189] mediates the oxidation of cysteine residues [190], causing allosteric changes within important signalling proteins and modifying their behaviour. Also, H_2_O_2_- can promote tyrosine phosphorylation by activating protein tyrosine kinases. Other evidence suggests that ROS signalling is required for the maintenance of tissues since it can activate cellular stress pathways to diminish tissue degeneration and promote healthy ageing [191]. Besides, ROS are also essential for stem cell differentiation, as observed in different in vivo models: murine hematopoietic stem cells in a mouse model with reduced ROS levels because lacking AKT1 and AKT2 showed compromised differentiation [192], while in Drosophila hematopoietic progenitors, increasing ROS triggers differentiation while decreasing ROS impairs it [193]. Experiments in humans also reported the importance of mitochondrial ROS in muscle differentiation and the differentiation into adipocytes of bone marrow mesenchymal stem cells [194,195]. Accordingly, reduced ROS levels decrease the regenerative capacity of neural stem cells and spermatogonial stem cells [196,197]. However, a few conflicting studies claimed that a rise in ROS might harm stem cell function [198,199], suggesting that there might be exceptions to the rule.

The ROS situation, therefore, is not black or white, but the type of ROS, their localisation, and their concentration collectively determine whether redox signalling or oxidative stress-induced damage occurs. Therefore, the problem in ageing and senescence is not ROS on its own, but a dysregulated and atypically high production of them, paired with redox dyshomeostasis. In conclusion, while aberrant ROS generation likely plays a role in age-related pathologies, antioxidant therapies will need to be carefully modulated to be both effective and not toxic. 

## 10. Mitochondrial Permeability Transition Pore

The mPTP is a transmembrane protein or complex that controls mitochondrial permeability and can induce cell death after various stresses, including oxidative stress, adenine nucleotide depletion, increased phosphate concentration, and high mitochondrial calcium [200]. The opening of the pore leads to mitochondrial swelling, uncontrolled diffusion of molecules under 1500 Da across the IMM, and sustained loss of mitochondrial membrane potential [162]. The permeability of the mitochondrial membranes is central in the decision between cell survival or death, and what type of cell death, since the activation of the pore has been associated with both apoptosis and necrosis [201]. While many potential structural components of the mPTP have been proposed, including ATP synthase, adenine nucleotide translocase (ANT), the outer membrane voltage-dependent anion channel (VDAC), and the phosphate carrier (PiC), the exact subunit composition of the pore is still debated [201]. Its regulatory pathways are also unclear. Cyclophilin D (CypD) is the only clearly described mPTP regulator: it controls mPTP opening by sensitising it to calcium, inorganic phosphate, and ROS, while other stimuli can activate mPTP opening via CypD-independent pathways [200,202]. 

A common feature of old tissues and senescent cells is an increased concentration of mitochondrial Ca^2+^ and ROS, two conditions that can directly stimulate the opening of the mPTP [203,204], and decreased levels of NAD^+^ [132,133,134], which results in low levels of SIRT3 activity and thereby high levels of the active, acetylated, form of CypD [205]. Together, these alterations (high Ca^2+^, high ROS and low NAD^+^) make the mitochondria of old and senescent cells highly prone to mPTP opening. Studies in muscles from aged humans and rats reported reduced mitochondrial calcium retention capacity and sensitisation of the mPTP opening, leading to apoptosis [64,206]. Similarly, enhanced susceptibility to mPTP opening during ageing was found in the brain [207,208], the liver [208,209], and lymphocytes [210]. The consequences of increased mPTP are the collapse of mitochondrial membrane potential, reduced mitochondrial respiratory function, the release of mitochondrial Ca^2+^ and cytochrome *c*, and enhanced ROS generation [202,211], all events that have been linked to ageing. On the contrary, our knowledge about mPTP opening in senescent cells remains speculative and its potential use as a therapeutic target needs to be investigated. 

In different degenerative diseases, typically associated with ageing, the inhibition of the mPTP has been tested as a protective strategy to preserve cell survival, showing encouraging results. In mouse models of Parkinson’s disease and amyloid lateral sclerosis, for example, the prevention of the pore opening by CypD depletion or pharmacological inhibitors of the mPTP showed delayed onset of disease and extended lifespan [212,213,214]. The same happened in ageing-related bone loss, where CypD knock-out mice showed enhanced resistance to osteoporosis [215]. 

In most diseases associated with cellular senescence, instead, the goal is the opposite, namely, to eliminate senescent cells. While an increase in mPTP activation could lead to extensive toxicity and damage also proliferative cells, a deeper understanding of the mPTP activity and regulation could offer new opportunities for intervention. Indeed, while pore activation has traditionally been considered a death sentence for the cell, additional evidence proved that the mPTP can open in two different ways: permanently or transiently. While the sustained opening leads to cell death, the temporary activation or “flicker” of the pore [216] can have beneficial effects or induce protective pathways. It can allow calcium, ROS or other molecules release or exchange between the mitochondrial matrix and the cytosol, activate rescue pathways, or act as a signalling event. Since Ca^2+^ and ROS levels are elevated in aged and senescent cells, a transient variation of the mitochondrial permeability could represent a weapon against overload and toxicity, while its inhibition could target a specific vulnerability of these cells and have a senolytic effect. Transient opening and moderate loss of mitochondrial membrane potential could also correlate with activation of the UPRmt, which contributes to health and longevity, as previously discussed [102,107], favouring cell survival. The mPTP has, therefore, two highly different functions, one that rescues the organelle and the cell in cases of moderate stress and one that condemns it when its condition is beyond repair. Future research on mPTP targeting strategies should keep this important difference in mind and, ideally, identify ways to discriminate between the two pathways.

## 11. Apoptosis

Apoptosis is the process that leads to a controlled and programmed cell death, which can occur as a response to various damages or stresses, such as DNA damage, oxidative stress, immune reactions and absence of certain growth factors, hormones and cytokines, or as a natural part of embryonic development and ageing [217]. The apoptotic program can be initiated by different triggers and follow different signalling pathways, which generally share the activation of initiator caspases (as caspase 8 and 9) and culminate in the activation of executioner caspases (as caspase 3 or 7) to finally induce DNA fragmentation, degradation of cytoskeletal and nuclear proteins, cross-linking of proteins, and formation of apoptotic bodies. Mitochondria are responsible for the initiation of a key apoptotic pathway, as a result of internal stress or damage signals, which consists of the mitochondrial outer membrane permeabilisation (MOMP), regulated by the Bcl-2 family of proteins. The Bcl-2 family includes both pro- and anti-apoptotic proteins, which balance their activity in cells as needed. Upon elevated stresses, the pro-apoptotic proteins BAX and BAK oligomerise at the OMM [218,219], where they induce the release of cytochrome *c* and other proteins from the intermembrane space into the cytosol [220]. Once in the cytosol, cytochrome *c* induces the downstream activation of apoptotic protease activating factor-1 (Apaf-1), caspase 9, and finally, caspase 3, starting the execution pathway [221,222]. To avoid the undue activation of this suicide pathway, another group of proteins, including Bcl-xL, Bcl-2, and Bcl-W, prevent the oligomerisation of BAX/BAK [223,224].

The positive influence of the anti-apoptotic side of this family on healthy ageing was confirmed by the fact that Bcl-xL was found overexpressed in centenarian subjects [223]. Moreover, in “old” fibroblasts with high passage number, the level of anti-apoptotic proteins increases dramatically after UV stress and favours the development of a senescent phenotype, while in UV-damaged “young” cells with low passage number, the pro-apoptotic pathway is preferred [225]. These results suggest that activation or upregulation of the Bcl-xL pathway could be a valid anti-ageing strategy, but it can also help preserve damaged cells in older tissue, which are more prone to develop a senescent phenotype. Since senescent cells’ survival depends more than their proliferative counterparts on the anti-apoptotic activity of these members of the Bcl-2 family, their downregulation or inhibition has been exploited for their senolytic effect. One of the few and most used senolytic agents available to date is, indeed, Navitoclax (ABT-263), which induces apoptosis of senescent cells by inhibition of Bcl-W and Bcl-xL [226,227]. Preliminary data from clinical trials using Navitoclax in combination with Ruxolitinib in patients with myelofibrosis [228,229] showed encouraging outcomes, but further studies are necessary to fully evaluate the potential of this novel combination. Finding additional senolytic targets or strategies to make the already existing ones more specific is, therefore, one of the most active areas in ageing research. 

## 12. Epigenetic Regulation

Although the DNA code in our cells carries the genetic information, the epigenome is responsible for the accessibility, stability and regulation of that valuable information, connecting the genotype with the phenotype [230,231]. Epigenetic changes are reversible, can be driven by external or internal influences, and represent a key mechanism behind cellular alterations during ageing. In mammals, older individuals tend to present with CpG hypomethylation, especially at repetitive DNA sequences [232,233,234]_,_ histone modification, chromatin remodelling, and disruption of non-coding RNA [235]. Interestingly, while the majority of expressed miRNAs decline in the brain of aged animal models and humans, a small subset of non-coding RNAs was found selectively upregulated [236,237]. Epigenetic changes are also crucial for the induction, progression, and maintenance of senescence. In senescent human fibroblasts, the accumulation of a distinct heterochromatic structure (senescence-associated heterochromatic foci or SAHF) has been reported, possibly as a consequence of persistent DNA damage, decreased histone chaperone protein production, and decreased histone biosynthesis [238,239]. Moreover, histone acetylation seems to influence this phenotype. Indeed, Histone deacetylase 4 (HDAC4) is downregulated in oncogene-induced senescence, leading to the upregulation of senescence-associated genes [240]. Chromatin remodelling events and histones modifications, moreover, can influence the expression of SASP components [241]. Thus, the manipulation of these mechanisms is a prime target in age- and senescence-delaying interventions.

Even if mitochondria have their own genome, most of their proteins are nuclear-encoded and they need to communicate with the surrounding cellular environment. For this reason, the bi-directional communication between these organelles and the nucleus is constant and can lead to epigenetic modifications. In fact, mitochondrial dysfunctions invoke a process known as mitochondria-to-nucleus retrograde response, able to regulate nuclear-encoded gene expression and cellular metabolism [242]. Mitochondria provide numerous co-substrates produced in the Krebs cycle that are required for epigenetic and transcriptional processes, such as histone modifications and chromatin remodelling [243]. Changes in metabolism and metabolites level during ageing and senescence can therefore contribute to epigenetic modifications. In addition, mitochondria act as redox sensors able to identify stressful conditions and react by shaping the chromatin to promote survival or trigger senescence [244]. As a consequence, amelioration of mitochondrial functions would have positive effects also on the epigenetic state of the cell. Promising results have already been shown in in vivo models such as *C. elegans* [107], where elevated levels of the Krebs cycle intermediate α-ketoglutarate induced DNA and histone demethylation via activation of two histone demethylases, JMJD3 and PHF8, resulting in the removal of repressive marks, the induction of UPRmt gene expression, and extended lifespan. Similarly, mitochondrial ROS activate the DNA-damage-sensing kinases, Tel1p and Rad35p, resulting in enhanced subtelomeric silencing via inactivation of Rph1p, a histone H3K36 demethylase of the jumonji family of enzymes [245]. This represents another process through which the maintenance of mitochondrial homeostasis and ROS production under a certain threshold can promote longevity. 

## 13. Transplant of Younger Mitochondria: Another Weapon against Ageing

So far, we have focused on identifying alterations happening in mitochondria in aged and senescent cells and how they can be exploited as potential targets for therapies. In aged cells, these strategies aim to reduce the damages that progressively accumulated in the “old” organelles over the years and improve their functions and structure. The majority of these alterations are shared by senescent cells and the targeting of these pathways can help eliminate them, reduce the SASP, or prevent the development of a senescent phenotype in the first place, when it is driven by mitochondrial dysfunctions. Recently, a less canonical idea to improve aged cells’ condition and reduce senescent cells accumulation in ageing-associated diseases has emerged: obtain cells with “younger” mitochondria through mitochondrial transplantation. 

Mitochondrial transplant (mtTP) consists in extracting “young” and healthy mitochondria, injecting them into a patient, and allowing them to be absorbed into the cells. While the development of this technique in a laboratory setting is still at its initial stage, mtTP between cells is not an alien concept and has even been observed naturally in vitro and in vivo. For example, astrocytes close to the site of a focally induced cerebral ischaemia can transfer mitochondria to neurons [246]. Additionally, mitochondria can be transferred between cells through tunnelling nanotubes, thin plasma membrane structures connecting cells and allowing intercellular transfers of organelles, various plasma membrane components, and cytoplasmic molecules [247,248], or packed in extracellular vesicles [249]. In a laboratory setting, mitochondrial transfer has been attempted in numerous ways: via cytosol transfer, mitochondrial injection in cells, or injection in the bloodstream of animal models [250]. Intra-arterial injection of viable and respiration-competent autologous mitochondria isolated from pigs’ skeletal muscle was successfully used in the treatment of acute kidney injury and cardiac ischemia/reperfusion [251,252]. Similar results were shown in the lungs, where mitochondria were delivered either intra-arterially in the pulmonary artery or by a nebulizer [253]. Pre-labelled mitochondria were found up to 4 weeks after injection in the studied tissues, but their maintained functionality was unclear [251]. More recent studies demonstrated that mitochondrial transplant significantly up-regulates the expression of the mitochondrial complex II subunit SDHB in the hippocampus of aged mice [254] and improves basal respiration and ATP production 48 h post-transplantation in rats’ cardiomyocytes [255]. The observed benefits on respiration, however, appeared lost a month after the treatment, suggesting a potential for this technique in the treatment of acute injuries more than chronic conditions. Further research, optimisation, and technological advances are therefore necessary to determine if mtTP will be applicable in the treatment of diseases in the future, including age-associated pathologies. 

## 14. Conclusions

Taken together, the evidence presented in this review shows that mitochondria dysfunctions have a close relationship with ageing and cellular senescence. Several mitochondrial pathways have already been taken into consideration as potential therapeutic targets for ageing-associated diseases, and promising compounds have been developed. Future research will have to answer numerous open questions including: is it possible to completely restore mitochondrial function in senescent and aged cells? Which age- or senescence-associated aspects are the primary drivers of mitochondrial dysfunction and vice-versa? Which ones are targetable therapeutically? Answering some of these questions could get us closer to healthy ageing, with countless medical, social and economic benefits. 

## Figures and Tables

**Figure 1 pharmaceutics-15-00352-f001:**
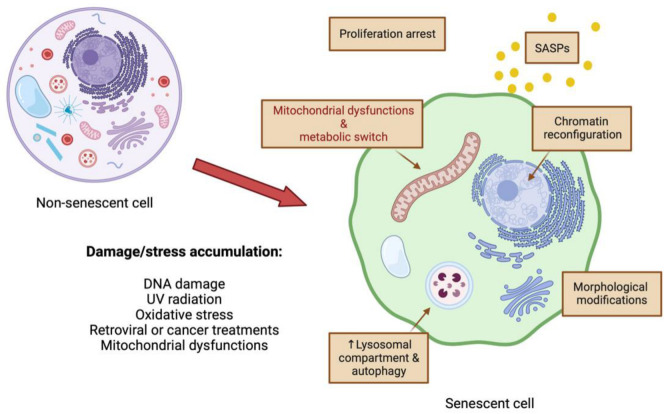
Cellular senescence is a terminal state of proliferation arrest in response to stressors or damages. Classic markers of cellular senescence are increased cell size, development of a senescence-associated secretory phenotype (SASP), chromatin remodelling, increase (↑) in the lysosomal compartment and autophagy, and mitochondrial alterations. Created with BioRender.com.

**Figure 2 pharmaceutics-15-00352-f002:**
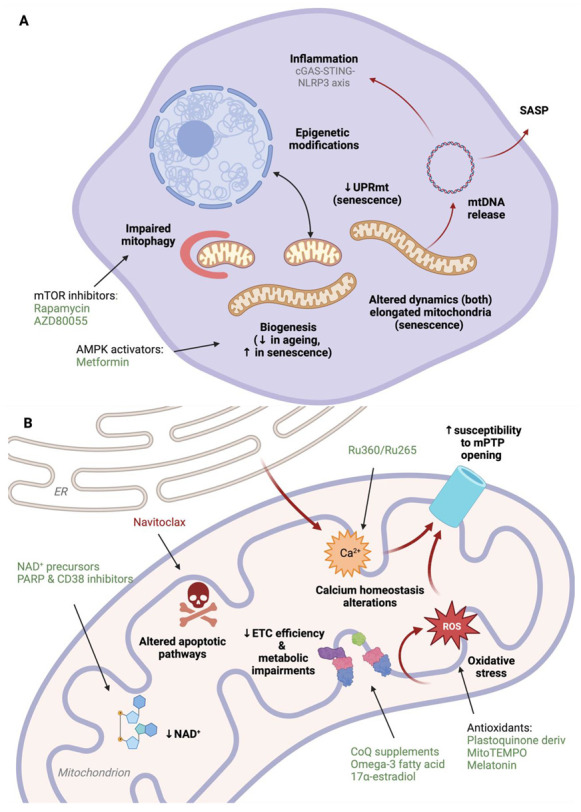
Mitochondrial dysfunctions are a hallmark of both ageing and cellular senescence and represent crucial targets in treating ageing-associated diseases. (**A**) Both aged and senescent cells are characterised by modifications in the mitochondrial network, dynamics, and interactions between the organelle and the rest of the cell. However, differences can be observed between the two conditions. In particular, modifications in the mitochondria-to-nucleus retrograde responses can induce different epigenetic changes in senescent versus old cells and modulate signal transcription pathways, such as the mitochondrial unfolded protein response (UPRmt), differently. Mitochondrial DNA (mtDNA) release, moreover, can be detected and cause inflammation in both conditions, but its role is particularly relevant as part of the senescence-associated secretory phenotype (SASP). (**B**) Most mitochondrial functions are similarly defective or altered in aged and senescent cells. Inefficient mitochondrial respiration results in impaired metabolism, a drop in NAD^+^ levels, and ROS production. Calcium and ROS accumulation might cause increased susceptibility to the mitochondria permeability transition pore (mPTP) opening and apoptosis. Arrows indicate increase (↑) and decrease (↓) of the correspondent feature. Created with BioRender.com.

## Data Availability

Data sharing not applicable.

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
