# Peer review of "Targeting Mitochondria to Control Ageing and Senescence"

_pharmaceutics, 2023, doi:10.3390/pharmaceutics15020352_

Round 1
Reviewer 1 Report
I have reviewed the paper entitled “Targeting mitochondria to control ageing and senescence”. Which need strong modification before acceptance.
Comments
-The abstract should be a summary of the entire text. The provided abstract is too much short, it should be more informative.
-the general introduction should be added before going to the subheading.
- all parts need to be updated by adding updated literature from SCOPUS.
-The references in the text are not cited properly.
The typographical and grammatical errors should be fixed.
Author Response
Reviewer 1
I have reviewed the paper entitled “Targeting mitochondria to control ageing and senescence”. Which need strong modification before acceptance.
We thank the reviewer for their comments. We have modified the manuscript according to their suggestions.
Comments
-The abstract should be a summary of the entire text. The provided abstract is too much short, it should be more informative.
We agree with the reviewer that a more informative abstract will be useful for readers approaching this manuscript. For this reason, the abstract has been now modified accordingly:
‘Ageing is accompanied by progressive impairment of cellular function and systemic deterioration of tissues and organs, resulting in increased vulnerability to multiple diseases. Here, we review the interplay between two hallmarks of ageing: mitochondrial dysfunction and cellular senescence. The targeting of specific mitochondrial features in senescent cells has the potential of delaying or even reverting the ageing process. A deeper and more comprehensive understanding of mitochondrial biology in senescent cells is necessary to effectively face this challenge. Here, we discuss the main alterations in mitochondrial functions and structure in both ageing and cellular senescence, highlighting the differences and similarities between the two processes. Moreover, we describe the treatments available to target these pathways and speculate on possible future directions of anti-ageing and anti-senescence therapies targeting mitochondria.’
-the general introduction should be added before going to the subheading.
We agree with the reviewer. Now the introduction does not have a subheading.
- all parts need to be updated by adding updated literature from SCOPUS.
We thank the reviewer for the suggestion. We have added the following updated references to the text (section, line; added text; [added reference]):
- Mitochondrial dynamics, line 226: The molecular mechanism of metformin is also debated, since metformin can activate AMPK as a downstream effect of its interaction with various proteins and pathways, including mitochondrial complex I, the nuclear receptor NR4A1, and the endosomal/lysosomal v-ATPase.
- [Zhou, J.; Massey, S.; Story, D.; Li, L. Metformin: An Old Drug with New Applications. Int J Mol Sci 2018, 19, 2863, doi:10.3390/ijms19102863.]
- [Triggle, C.R.; Mohammed, I.; Bshesh, K.; Marei, I.; Ye, K.; Ding, H.; MacDonald, R.; Hollenberg, M.D.; Hill, M.A. Metformin: Is It a Drug for All Reasons and Diseases? Metabolism 2022, 133, 155223, doi:10.1016/j.metabol.2022.155223.].
- UPRmt, line 271. Similarly, experiments in senescent human lung fibroblasts showed a reduced ability to cope with the accumulation of mitochondrial unfolded proteins.
- [Lee, T.-Y.; Huang, L.-J.; Dong, H.-P.; Tohru, Y.; Liu, B.-H.; Yang, R.-C. Impairment of Mitochondrial Unfolded Protein Response Contribute to Resistance Declination of H2 O2 -Induced Injury in Senescent MRC-5 Cell Model. Kaohsiung J Med Sci2020, 36, 89–97, doi:10.1002/kjm2.12146.]
- Metabolism, line 302: CoQ10 supplementation could have beneficial effects also against the development of cellular senescence, as demonstrated in mesenchymal stem cells and H2O2-induced senescent HUVECs.
- [Huo, J.; Xu, Z.; Hosoe, K.; Kubo, H.; Miyahara, H.; Dai, J.; Mori, M.; Sawashita, J.; Higuchi, K. Coenzyme Q10 Prevents Senescence and Dysfunction Caused by Oxidative Stress in Vascular Endothelial Cells. Oxid Med Cell Longev 2018, 2018, 3181759, doi:10.1155/2018/3181759.]
- SIRT3 modulation, NAD+ levels, line 378: …adjudin, a derivative of lonidamine, which was described attenuating cellular senescence markers in hydroxyurea-treated MEFs.
- [Geng, K.; Fu, N.; Yang, X.; Xia, W. Adjudin Delays Cellular Senescence through Sirt3‑mediated Attenuation of ROS Production. Int J Mol Med 2018, 42, 3522–3529, doi:10.3892/ijmm.2018.3917.].
- Apoptosis, line 570: Preliminary data from clinical trials using Navitoclax in combination with Ruxolitinib in patients with myelofibrosis showed encouraging outcomes, but further studies are necessary to fully evaluate the potential of this novel combination.
- [Pemmaraju, N.; Garcia, J.S.; Potluri, J.; Harb, J.G.; Sun, Y.; Jung, P.; Qin, Q.Q.; Tantravahi, S.K.; Verstovsek, S.; Harrison, C. Addition of Navitoclax to Ongoing Ruxolitinib Treatment in Patients with Myelofibrosis (REFINE): A Post-Hoc Analysis of Molecular Biomarkers in a Phase 2 Study. Lancet Haematol 2022, 9, e434–e444, doi:10.1016/S2352-3026(22)00116-8.223. ]
- [Harrison, C.N.; Garcia, J.S.; Somervaille, T.C.P.; Foran, J.M.; Verstovsek, S.; Jamieson, C.; Mesa, R.; Ritchie, E.K.; Tantravahi, S.K.; Vachhani, P.; et al. Addition of Navitoclax to Ongoing Ruxolitinib Therapy for Patients With Myelofibrosis With Progression or Suboptimal Response: Phase II Safety and Efficacy. J Clin Oncol 2022, 40, 1671–1680, doi:10.1200/JCO.21.02188.]
-The references in the text are not cited properly.
All the references have been added using the reference manager Zotero and the Pharmacuetics specific citation style has been used. All the references have been checked and updated in order to assure the correct format.
The typographical and grammatical errors should be fixed.
The text has been revised both manually and using the software Grammarly and all the detected typos and errors have been corrected.
Reviewer 2 Report
This is an exciting and timely review of mitochondria as a target for improving health in aging and senescence. Many of the aspects described in the manuscript could be relevant to other aspects of health/disease pertinent to cancer, metabolic syndromes, genetic disorders, etc. One possible improvement is adding more figures or illustrations (at the moment, there is only one figure in the paper).
Author Response
Reviewer 2
This is an exciting and timely review of mitochondria as a target for improving health in aging and senescence. Many of the aspects described in the manuscript could be relevant to other aspects of health/disease pertinent to cancer, metabolic syndromes, genetic disorders, etc. One possible improvement is adding more figures or illustrations (at the moment, there is only one figure in the paper).
We thank the reviewer for the comments. We have added a figure (now Figure 1) to better explain the characteristics and the main markers of senescent cells.
Reviewer 3 Report
This review article is well written, and the authors have explained very clearly the mitochondrial dysfunction in aging and senescence.
There are some minor suggestions:
1. Line 31, 32: Add reference for the statement “The words senescence and aging………scientists use them indistinguishably”.
2. Line 229 – 232: Increased AMPK activity (Ref 86) and reduced AMPK activity (Ref 87) were observed in senescent cells. Were these both studies done on similar cells/tissues? Was the activity of AMPK differing in a tissue-specific manner or does the AMPK activity differ with various mode of senescence induction?
3. Section 8: Metabolism and electron transport chain: A little information can be provided about how the senescent cells shift their metabolism while their ETC efficiency and ATP production were reduced.
4. Line 327: PARP’s role in senescence – were these contradictory roles reported in the same or different cell lines or tissues? And were these reported in the different senescent induction? Kindly give the details.
Author Response
Reviewer 3
This review article is well written, and the authors have explained very clearly the mitochondrial dysfunction in aging and senescence.
We thank the reviewer for their comment.
There are some minor suggestions:
- Line 31, 32: Add reference for the statement “The words senescence and aging………scientists use them indistinguishably”.
We appreciate this comment. The distinction between “cellular senescence” and “cellular ageing” is a personal choice of the authors to help clarify two cellular processes that are related but not exactly identical. To make clear that this is our personal position, we have deleted the first sentence.
- Line 229 – 232: Increased AMPK activity (Ref 86) and reduced AMPK activity (Ref 87) were observed in senescent cells. Were theseboth studies done on similar cells/tissues? Was the activity of AMPK differing in a tissue-specific manner or does the AMPK activitydiffer with various mode of senescence induction?
Thank you for this clarification. The two models are indeed different, AMPK increase was observed in human fibroblasts while AMPKdecrease was observed in murine fibroblasts. In the first case, immortalized fibroblasts expressing SV40 large T were rendered senescent by repression of large T (current reference 88). In the second study, senescence was induced by H2O2 treatment of murine fibroblasts (current reference 89).
The text has been modified accordingly:
‘Interestingly, the role of AMPK activity and its function in the regulation of mitochondrial biogenesis in cellular senescence is less clear. A study in immortalised human fibroblasts expressing SV40 large T antigen showed that inactivation of large T resulted in an increase in AMPK activity that directly contributed to the implementation of senescence [88]. In contrast, a study in H2O2-induced senescent murine fibroblasts found AMPK inactivated [89]. Since AMPK is central in multiple cellular pathways, additional research could clarify the observed differences between models, and unravel additional molecular mechanisms involved in the establishment and/or maintenance of senescence.’ [see lines 234-241]
- Section 8: Metabolism and electron transport chain: A little information can be provided about how the senescent cells shift their metabolism while their ETC efficiency and ATP production were reduced.
The following text and references have been added:
‘Because of this defect, senescent cells appear to undergo a metabolic switch, increasing their dependence on glycolysis [111], and fatty acid oxidation [112].’ [see lines 284-286]
- James, E.L.; Michalek, R.D.; Pitiyage, G.N.; de Castro, A.M.; Vignola, K.S.; Jones, J.; Mohney, R.P.; Karoly, E.D.; Prime, S.S.; Parkinson, E.K. Senescent Human Fibroblasts Show Increased Glycolysis and Redox Homeostasis with Extracellular Metabolomes That Overlap with Those of Irreparable DNA Damage, Aging, and Disease. J Proteome Res 2015, 14, 1854–1871, doi:10.1021/pr501221g.
- Quijano, C.; Cao, L.; Fergusson, M.M.; Romero, H.; Liu, J.; Gutkind, S.; Rovira, I.I.; Mohney, R.P.; Karoly, E.D.; Finkel, T. Oncogene-Induced Senescence Results in Marked Metabolic and Bioenergetic Alterations. Cell Cycle 2012, 11, 1383–1392, doi:10.4161/cc.19800.
- Line 327: PARP’s role in senescence – were these contradictory roles reported in the same or different cell lines or tissues? And were these reported in the different senescent induction? Kindly give the details.
Thank you for the comment. In this case, all the references used described experiments in human cells (keratinocytes, ovarian cancer, colorectal cancer, melanoma and breast cancer). However, the methods used to induce senescence were different, so the following sentence has been added to avoid confusion.
‘While all the mentioned studies have been conducted on human cells, the use of different cell models (keratinocytes, ovarian cancer, colorectal cancer, melanoma and breast cancer cells) and the use of different senescence triggers might explain these disparities.’ [see lines 345-348]
Round 2
Reviewer 1 Report
The paper should be accepted.